# Curcumin Induces Homologous Recombination Deficiency by BRCA2 Degradation in Breast Cancer and Normal Cells

**DOI:** 10.3390/cancers17132109

**Published:** 2025-06-24

**Authors:** Zofia M. Komar, Marjolijn M. Ladan, Nicole S. Verkaik, Ahmed Dahmani, Elodie Montaudon, Elisabetta Marangoni, Roland Kanaar, Julie Nonnekens, Adriaan B. Houtsmuller, Agnes Jager, Dik C. van Gent

**Affiliations:** 1Department of Molecular Genetics, Erasmus Medical Center Cancer Institute, University Medical Center, 3015CN Rotterdam, The Netherlands; 2Oncode Institute, 3521 Utrecht, The Netherlands; 3Translational Research Department, Institut Curie, Paris Sciences et Lettres University, 75005 Paris, France; 4Department of Radiology and Nuclear Medicine, Erasmus Medical Center Cancer Institute, University Medical Center, 3015CN Rotterdam, The Netherlands; 5Erasmus Optical Imaging Center and Department of Pathology, Erasmus Medical Center, 3015CN Rotterdam, The Netherlands; 6Department of Medical Oncology, Erasmus Medical Center Cancer Institute, University Medical Center, 3015CN Rotterdam, The Netherlands

**Keywords:** curcumin, PARPi, breast cancer

## Abstract

Many breast cancer patients seek supplementation on top of the conventional anti-cancer treatment, in the hope that it will help shrink their tumors. Curcumin is an example of such a supplement, often used by cancer patients due to the wide range of its health claims. Interestingly, recent research has shown that curcumin can also affect an important DNA repair pathway—homologous recombination (HR). Therefore, our study aimed to determine the potential effect of curcumin combined with a breast cancer therapy that is effective in HR deficient tumors, Poly (ADP-ribose) polymerase inhibitor (PARPi). We showed that curcumin treatment sensitized not only the cancerous but also the normal cells to PARPi treatment, suggesting that curcumin supplementation in patients undergoing PARPi therapy should be approached with caution.

## 1. Introduction

Breast cancer (BC) is the most common cancer in women worldwide [1]. Despite efforts to improve early detection and therapy, BC remains among the deadliest cancers, with 685,000 deaths in 2020 [2]. Approximately 20–30% of these patients use herbal supplements alongside treatment [3], making their effects worth investigating. A commonly used compound is curcumin, derived from turmeric (*Curcuma longa*), long used in Traditional Chinese Medicine for its anti-inflammatory properties [4]. Curcumin has shown potential in enhancing anti-cancer treatment for colorectal [5], pancreatic [6], cervical [7], breast [8], and lung cancer [9], and leukemia [10]. As a result, curcumin food supplementation is broadly advertised on patient forums and websites as beneficial for anti-cancer treatment.

Curcumin is sold as an over-the-counter (OTC) drug at relatively high doses compared to dietary intake. Moreover, new curcumin formulations (e.g., with piperine) enhance bioavailability, further increasing blood plasma concentrations [11]. OTC curcumin may significantly influence patients—positively or negatively. Three phase I clinical studies have assessed curcumin monotherapy tolerance and serum levels after administration [12,13,14]. In two cases using regular curcumin formulation, the maximal serum levels reached in patients were relatively low [13,14]. In the study of Sharma et al., the effectiveness of the treatment was limited—two out of fifteen patients involved in the study had progressive disease before and a stable disease after the curcumin treatment. In Greil et al.’s study, a highly bioavailable curcumin (Lipocurc™) was tested in a dose-escalation trial. While the initial dose caused no toxicity, no radiological response was observed. Shortening the administration interval led to hemolysis in one of six patients and a serious hemoglobin drop (>2 g/L) in three others. This suggests that non-toxic curcumin levels in the body are not sufficient to provide anti-tumor effects. However, recent preclinical studies indicate that lower doses may enhance sensitivity to other anti-cancer treatments [15].

BC is a highly heterogeneous type of cancer that can occur due to inherited factors or spontaneous genetic aberrations. Hereditary BC is most often caused by mutations in the breast cancer 1 (*BRCA1*) or breast cancer 2 (*BRCA2*) genes [16]. Inactivating mutations in the *BRCA1* and *BRCA2* genes lead to a deficiency in the Homologous Recombination (HR) pathway. This DNA repair pathway accurately resolves double-stranded breaks (DSBs) by using the sister chromatid as a template [17]. Poly (ADP-ribose) polymerase inhibitor (PARPi) treatment specifically targets HR deficient (HRD) tumors. This treatment offers several advantages compared to other available therapies, such as lower side effects and convenient administration in a tablet form [18]. Nevertheless, only a minority of all BCs have an HRD phenotype and can benefit from treatment with PARPi. Recently, various methods to potentially induce the HRD phenotype in non-HRD cancer cells have been described, including hyperthermia treatment (42 °C) [19], aldehydes [20], bortezomib [21], B-thujaplicin [22], caffeine [23], and curcumin [15].

In the study by Ogiwara et al. (2013) [15], curcumin-induced HRD was demonstrated by a reduction in RAD51 ionizing radiation-induced focus (IRIF) formation. However, since RAD51 IRIF only forms during the S/G2 phase of the cell cycle [24], the possibility of cell cycle arrest contributing to this observation could not be ruled out. For that reason, we validated the HR defect by showing the inhibitory effect of curcumin treatment in replicating cells specifically. Furthermore, this study aimed to investigate the sensitizing potential of curcumin to PARPi treatment in both cancerous and normal cells.

## 2. Materials and Methods

### 2.1. Cell Culture

Eight BC cell lines and three normal cell lines with confirmed identity and known origin were cultured with 5% CO_2_ at 37 °C. All media were supplemented with 10% fetal bovine serum (FBS; Bondico BV, Alkmaar, The Netherlands)—except 15% FBS for the C5RO cell line; and 1% penicillin streptomycin (Sigma-Aldrich, St. Louis, MO, USA). Cell lines were culture in following medium: T47D (RRID: CVCL_0553), HCC1937 (RRID: CVCL_0290) and MM436 (MDA-MB-436; RRID: CVCL_0623) in Roswell Park Memorial Institute (RPMI) 1640 (1X) + GlutaMAX™ (Gibco, Thermo Fisher Scientific, Waltham, MA, USA); Sum149 (RRID: CVCL_3422), Sum44 (RRID: CVCL_3424), HCC1143 (RRID: CVCL_1245), ZR.75.1 (RRID: CVCL_0588) and MCF7 (RRID: CVCL_0031) in RPMI-1640 medium (Sigma-Aldrich, St. Louis, MO, USA); HUVEC cells in the EGM™-2 Endothelial Cell Growth Medium-2 Bullet Kit™ (CC3162; Lonza, Basel, Switzerland); C5RO cells (RRID: CVCL_ZP35) in the Ham/F10 medium (P04-12500; PAN-Biotech, Aidenbach, Germany) and MCF10a cells (RRID: CVCL_0598) in DMEM/F12 medium (11965-118; Invitrogen, Thermo Fisher Scientific, Waltham, MA, USA) supplemented as previously described [25]. All BrC cell lines were kindly provided by the Department of Medical Oncology (Erasmus MC), previously described [26]. HUVEC cells were purchased through Innoprot (P20201; Innoprot, Derio, Spain). The remaining cell lines were kindly provided by colleagues from the Department of Molecular Genetics at the Erasmus MC. The HeLa cell line is widely used in different laboratories in various forms and is highly divergent in the number of chromosomes. In this study, a line commonly used in the department was used without further characterization. The main goal of this paper was to show the broad spectrum of curcumin’s effects on various cells, making the specification of each cell line less relevant.

### 2.2. PDX Tissue Handling

Patient-derived xenografts (PDXs) of BC were established at the Institut Curie in Paris, as described previously [27,28]. BC PDX tissue samples were collected and processed as previously described [29]. PDX models were used in this study solely to validate the findings from cell lines using a 3D tissue model, and no further experiments on animals were performed.

### 2.3. Cell and Tissue Treatment

BC cell lines and PDX models were treated with curcumin 5–50 µM curcumin (diluted in DMSO and stored at −20 °C; Sigma-Aldrich^®^, St. Louis, MO, USA; based on previous studies [15,30]). BC and normal cells were treated with curcumin combined with olaparib (Selleck Chemicals LLC, Houston, TX, USA) at a concentration range of 0.1–10 µM. To visualize replicating cells, 30 µM 5-Ethynyl-2′-deoxyuridine (EdU) was added to the cells and the tissue slices, 30 min and 2 h prior to fixation, respectively. To investigate the mechanism of protein degradation, the following compounds were used for the treatment of cell lines: MG132 (EMD Millipore Corp., Billerica, MA, USA), N-acetylcysteine (NAC; diluted in DMSO; Sigma-Aldrich^®^, St. Louis, MO, USA), NMS-873 (Selleck Chemicals, Houston, TX, USA), 1G244 (AK Scientific, Ahern Avenue Union City, CA, USA), chloroquine (Sigma-Aldrich^®^, St. Louis, MO, USA) and ganetespib (S1159; Selleck Chemicals, Houston, TX, USA).

### 2.4. Irradiation

Cells and BC PDX tissue slices were irradiated 2 h post curcumin treatment with an RS320 X-ray machine, at a dose rate of 0.9 Gy/min, current of 10 mA, voltage of 195 kV, and the 0.5 mm Cu filter (Xstrahl Ltd., Live Sciences, Suwanee, GA, USA). Following irradiation, the cells were incubated for 2 h to allow the formation of RAD51 IRIF.

### 2.5. Immunostaining

Cells seeded on coverslips were immunostained for RAD51 IRIF in combination with a Click-iT reaction (EdU visualization). The following antibodies were used: primary rabbit anti-RAD51 (1:10,000) [31] and anti-53BP1 (Novus Biologicals^®^, Littleton, CO, USA), and a secondary goat-anti-rabbit Alexa-Fluor-594 (Thermofisher Scientific, Waltham, MA, USA). The Click-iT reaction was performed using Atto 488 azide (Atto-tec GmbH, Sieden, Germany). All nuclei were visualized using Vectashield containing DAPI (Vector Laboratories, Inc., Burlingame, CA, USA). The RAD51/Geminin staining on PDX tissue slices was performed as described previously [24].

### 2.6. Imaging Acquisition and Analysis

All representative images were acquired with a Leica-SP5 confocal microscope (Leica Microsystems, Wetzlar, Germany) using a 63× objective. To calculate the RAD51 IRIF positive cells, the Leica DM4000 B (Leica Microsystems, Wetzlar, Germany) with a 60× objective was used. RAD51 IRIF was only counted in EdU-positive cells. A cell was considered RAD51 IRIF positive if ≥5 foci could be observed. At least 30, but preferably around 100, EdU-positive nuclei were counted in each sample for an accurate result.

### 2.7. Western Blotting

Western Blot was performed as previously described [32]. The following antibodies were used: DNA-PK_CS_ (home-made, 1:1000) [33], BRCA2 (OP95; EMD Millipore, Burlington, MA, USA, 1:1000), PARP1 (BML-SA249-0050; Enzo Life Sciences, Inc., Farmingdale, NY, USA, 1:2000), HSP70 (4873S; Cell Signaling Technology, Danvers, MA, USA), α-tubulin (05-829; Merck KGaA, Darmstadt, Germany, 1:5000) and anti-actin clone C4 (mab1501r; Millipore, Burlington, MA, USA, 1:50,000) and secondary HRP-labeled Sheep-α-mouse (Jackson Immuno Research, West Grove, PA, USA, 1:1000) or Donkey-α-rabbit (Jackson Immuno Research, West Grove, PA, USA, 1:1000). Quantification of each band was performed using ImageJ (version 1.54i), measuring the intensity of each band as described previously [34].

### 2.8. Clonogenic Survival Assay

Cells were seeded on 6-well plates at low density and incubated overnight to allow attachment. Next, cells were treated short-term with curcumin (2.5 h; Appendix A) and/or constantly with olaparib and subsequently incubated to allow colony formation.

### 2.9. Statistics

All statistical analyses were conducted using GraphPad Prism (version 9). A standard one-way ANOVA test was used, combined with Tukey’s multiple comparisons test to investigate potential significant differences. *p*-values were considered significant below 0.05.

## 3. Results

### 3.1. Curcumin Treatment Inhibits RAD51 IRIF Formation in Replicating Cells

The loss of RAD51 IRIF upon curcumin treatment was shown previously for multiple cancer cell lines [15]. To validate the observed HRD induction, we assessed the RAD51 IRIF formation capacity in replicating cells specifically, using EdU as an S-phase-specific cell cycle marker. Curcumin treatment led to a reduction in the fraction of replicating cells that were RAD51 IRIF positive in a dose-dependent manner for all tested HRP cell lines (Figure 1A–C). The significant reduction of RAD51 IRIF-positive cells was shown at a relatively low curcumin concentration of 10 μM for most HRP cell lines. After treatment with 30 μM curcumin, all RAD51 IRIF formation was completely inhibited in BC cell lines. Curcumin treatment selectively reduced RAD51 IRIF without significantly affecting 53BP1 IRIF formation (Appendix A), suggesting the specificity of the effect.

In normal cells (Figure 1D), treatment with 30 µM curcumin resulted in a significant reduction of RAD5 IRIF in endothelial (HUVEC; *n* = 1) and skin fibroblasts (C5RO; *n* = 1). The normal breast epithelium cell line (MCF10a; *n* = 3) showed less severe reduction of RAD51 IRIF at 30 µM curcumin. However, 50 µM curcumin completely abrogated RAD51 IRIF formation.

This same effect of curcumin was further validated in BC tumors using a PDX model (proven to be HRP by the RECAP assay [24]), showing reduction in RAD51 IRIF numbers in a dose-dependent manner (Figure 2A,B). We conclude that HRD induction is not an artefact of 2D cell cultures.

### 3.2. RAD51 IRIF Loss Is Caused by Curcumin-Induced BRCA2 Degradation

RAD51 accumulation at DSBs is mediated by the BRCA2 protein [35]. Therefore, we investigated the effect of curcumin treatment on the levels of BRCA2 protein. We found a dose-dependent degradation of BRCA2 (Figure 3A–D), but not RAD51 protein (Figure 3E,F).

The effect of curcumin on the BRCA2 levels was transient; cells that were incubated for 4, 24, and 48 h after the 2 h long curcumin treatment showed recovery of BRCA2 to the initial levels at 24 h after washing the curcumin away (Figure 4A,B). To better understand the effect of curcumin on protein levels in BC cells, we investigated its influence on the heat shock protein 90 (HSP90). HSP90 is a chaperone protein that aids protein folding, and changes in its activity could potentially affect the stability of various proteins [36]. Curcumin treatment for 48 h resulted in increased HSP70 levels, indirectly marking HSP90 inhibition (Figure 4C,D [37]). To further investigate the mechanism of curcumin-induced BRCA2 degradation, different pathways were investigated, including the following: (i) the proteasome, (ii) oxidative stress induction, (iii) autophagy, (iv) p97-mediated degradation, and (v) dipeptidyl peptidase 9 (DDP9) mediated degradation (Appendix A). BRCA2 levels were not rescued after treatment with any of the tested inhibitors, suggesting that another mechanism is involved or that at least two redundant pathways exist for this degradation.

### 3.3. Curcumin Treatment Specifically Induces HRD and Sensitizes HRP Cell Lines to PARPi Therapy

To determine whether the observed reduction in RAD51 IRIF after curcumin treatment is directly associated with HRD induction, we investigated whether curcumin treatment can sensitize cancerous cells to PARPi treatment. For this purpose, T47D (BC cell line, *BRCA1* and *BRCA2* wt., HRP) and HeLa cells (cervical cancer cell line, *BRCA1* and *BRCA2* wt., HRP) were treated with a combination of curcumin (10, 20, and 30 µM) and the PARPi olaparib (0.1 and 0.25 µM). These curcumin concentrations hardly reduce the viability of the cells (Figure 5F). However, it sensitized both cell lines to olaparib treatment in a dose-dependent manner (Figure 5A,B). Treatment with curcumin also sensitized the MCF10a cell line to olaparib in a dose-dependent manner (Figure 5C), showing that curcumin treatment can also affect normal cell viability upon PARPi treatment. To determine whether this sensitization was specific for olaparib treatment, a colony survival assay was performed after the combination treatment of curcumin and ionizing radiation. As damage induced by ionizing radiation is mainly repaired by non-homologous end-joining and not HR [38], curcumin should not induce much increased sensitivity to the treatment. Here we show that curcumin treatment indeed did not sensitize the T47D and HeLa cell lines to irradiation treatment (Figure 5D–F). This suggests that curcumin has an inhibitory effect on HR specifically and does not affect NHEJ.

## 4. Discussion

In this study, the effect of curcumin on the HR pathway was investigated. We showed that curcumin treatment caused a reduction of RAD51 IRIF and BRCA2 protein levels, resulting in sensitization to PARPi treatment. This effect has been shown for both cancerous and non-cancerous cell lines, suggesting that even though efficacy can be improved, combination treatment may also lead to increased PARPi toxicity.

The first part of this study assessed the specificity of the HRD induction observed in curcumin-treated cell lines. Multiple control experiments were conducted, confirming the specificity of RAD51 IRIF reduction, BRCA2 degradation, and PARPi sensitization. Nevertheless, the exact mechanism by which BRCA2 is degraded following curcumin treatment remains unclear. We hypothesize that curcumin’s effects are broad, extending beyond its effect on DNA repair pathways, which complicates the identification of a single mechanism responsible for BRCA2 degradation. This hypothesis can be supported by the observed curcumin-induced HSP90 inhibition shown in this study, consistent with previous findings [39,40]. The effect of curcumin on HSP90 was similar to the inhibition after treatment with the known second-generation HSP90 inhibitor, ganetespib [41]. Notably, another study found very comparable effects of ganetespib treatment to those of curcumin presented in our study, including RAD51 IRIF reduction, BRCA2 degradation, and sensitization to PARPi treatment [42]. This further validates our hypothesis that curcumin affects a broad spectrum of various pathways involved in protein stability by inhibiting the HSP90 protein activity, which could also be the cause for the observed BRCA2 degradation. Various studies have confirmed this, pointing out curcumin as a PAIN (pan assay interference) compound, a name referring to substances that do not have a specific target, but affect the function of diverse pathways and different diseases [43,44]. This suggests that curcumin treatment is not specific to one mechanism and can have various, even very subtle effects on both the cancer and normal cells within the body. Considering this, administration of higher curcumin concentrations should be taken with caution as long as its exact effects are not well defined.

Multiple in vitro studies have shown that curcumin can inhibit proliferation and induce apoptosis in various tumor types [45,46,47] and therefore potentiate the response to various treatments [48,49,50,51]. However, many studies did not consider the potential side effects on normal cells that can be induced by curcumin treatment. One study showed that curcumin treatment can induce cell death in HUVEC [30]. In contrast, several other studies showed curcumin induced cell death specifically in tumor cells, with no toxicity in normal cell lines [52,53,54]. Our study shows that curcumin treatment with concentrations reaching 30 µM for 2.5 h results in a slight decline of colony-forming capabilities, but no differences between the two cancerous and one normal cell line were detected. Even though limited toxicity is induced, a clear effect on the RAD51 IRIF and BRCA2 protein levels was shown in the normal breast epithelial cell line (MCF10a) and the tested cancer cell lines. Moreover, the combination treatment of curcumin and PARPi sensitized both the normal and cancerous cell lines to PARPi treatment. This shows that the absence of direct cell death induction after curcumin monotherapy in the normal and cancerous cells in vitro does not mean that these cells are not affected by the treatment.

After analyzing the effect of curcumin on both normal and cancerous cells in our own and other in vitro studies, we considered whether these effects could have been expected in patients. Regular formulations of curcumin have low bioavailability [55], and the concentrations reached in patients in vivo are relatively low compared to those achieved in the in vitro studies [56]. Nevertheless, a study performed by Hussaarts et al. demonstrated that curcumin affected tamoxifen pharmacokinetics, suggesting that even low levels of curcumin can significantly impact patients [57]. Two phase I clinical studies have been performed to determine the toxicity of regular curcumin formulations in patients. One study showed that 24 h after daily administration of 8 g curcumin, the peak serum concentration reached approximately 1.77 µM [14]. Patients tolerated the tablets well, i.e., after three months on treatment, no dose-limiting toxicity (above grade 2) was mentioned amongst the patients in the trial (*n* = 25). Further dose escalation to 12 g a day was not possible because the pills were too large to swallow easily. Another study showed lower serum peak concentrations of curcumin one hour after oral administration of 3.6 g of curcumin, reaching 0.011 µM [13]. In this study, again, no severe toxicity was found after four months of daily treatment among 15 patients. The only adverse event described was mild diarrhea.

To overcome the low bioavailability of curcumin, new formulations of the supplements have been developed, including nano-formulations, liposomes, and micelles [58]. A phase I clinical study investigated the effect of liposomal curcumin (Lipocurc™), which is a newly developed form of curcumin that increases the bioavailability of the compound [12]. Here, a weekly dose of liposomal curcumin of 100–300 mg/m^2^ administered intravenously during eight hours showed no dose-limiting toxicity in 32 tested patients. However, no radiological response to the curcumin monotherapy was registered between 4 and 8 weeks of treatment. An attempt to shorten the administration duration (300 mg/m^2^ in six hours) failed as this led to hemolysis in one of the six patients and a serious fall in hemoglobin (>2 g/L) in three other patients, underscoring that curcumin supplementation may lead to severe side effects. The maximal serum concentration of curcumin reached in the patients was 1400 ng/mL (0.5 µM). These results suggest that the higher levels used in vitro are not easily reached in patients, but this might change with novel ways of curcumin administration.

## 5. Conclusions

In conclusion, we showed that the curcumin-induced HRD phenotype could sensitize both cancerous and non-cancerous cells to PARPi treatment. Considering the variable scientific evidence on normal cell survival after curcumin treatment, more research is required to determine the effect of curcumin on different tissue types, and potential combination treatments in patients should be approached carefully. Clinical studies evaluating plasma levels of regular curcumin supplementation showed good tolerance in patients, but the maximum concentration reached in blood was much lower than the concentration at which effects were observed in our in vitro experiments. Results from clinical studies, including treatment with the Lipocurc^®^ formulation [12], suggest that curcumin treatment may carry potential risks, particularly with formulations that result in higher curcumin concentrations. For that reason, the clinical use of curcumin, especially when combined with DNA-damaging agents, should be approached with caution.

## Figures and Tables

**Figure 1 cancers-17-02109-f001:**
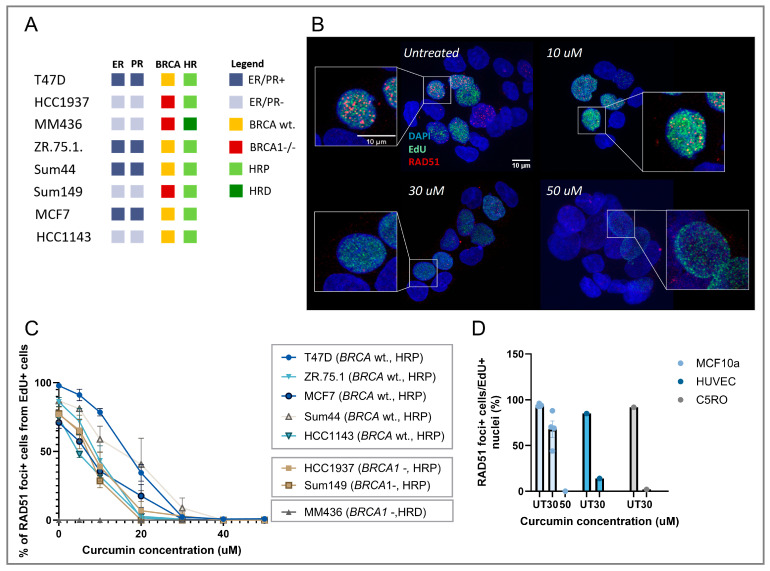
Effects of curcumin treatment on RAD51 IRIF formation in cells in S-phase. (**A**) Overview of eight BC cell lines representing different types of BC based on ER/PR, BRCA1/2, and HR status used for the RAD51 IRIF formation assay. (**B**) Representative microscope images of RAD51 IRIF after curcumin treatment, with blue channel representing DAPI, green EdU, and red RAD51 IRIF signal. (**C**) Quantification of the RAD51 IRIF positive cells out of the total of S-phase cells after curcumin treatment with a dose range between 5 and 50 µM. Each data point represents the read-out per cell line for each replicate separately, *n* = 3. (**D**) Quantification of the RAD51 IRIF positive cells out of all S-phase cells after 30 µM curcumin treatment in three different normal cells of the epithelium (MCF10a; *n* = 3), endothelium (HUVEC; *n* = 1), and skin fibroblasts (C5RO; *n* = 1).

**Figure 2 cancers-17-02109-f002:**
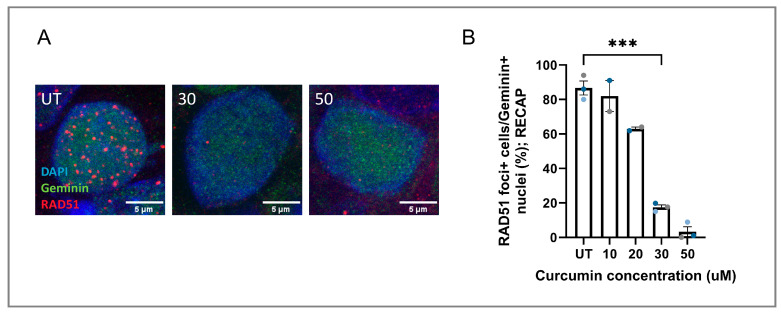
Effects of curcumin treatment on RAD51 IRIF in BC PDX tissue slices. (**A**) Representative microscope images of RAD51 IRIF in the RECAP assay, before and after curcumin treatment in the BC PDX-derived tissue slices. (**B**) Quantification of the RECAP assay read-out, where RAD51 IRIF-positive nuclei are calculated from the total of geminin-positive nuclei (*n* = 3; except for 10 and 20 µM, where treatment was performed for two PDX tumors). Each different dot color represents the read-out from a separate PDX tumor. Significant differences are indicated in the graph based on a standard one-way ANOVA test. Asterisks represent the significance based on the *p*-value, as follows: triple (***) *p* ≤ 0.001.

**Figure 3 cancers-17-02109-f003:**
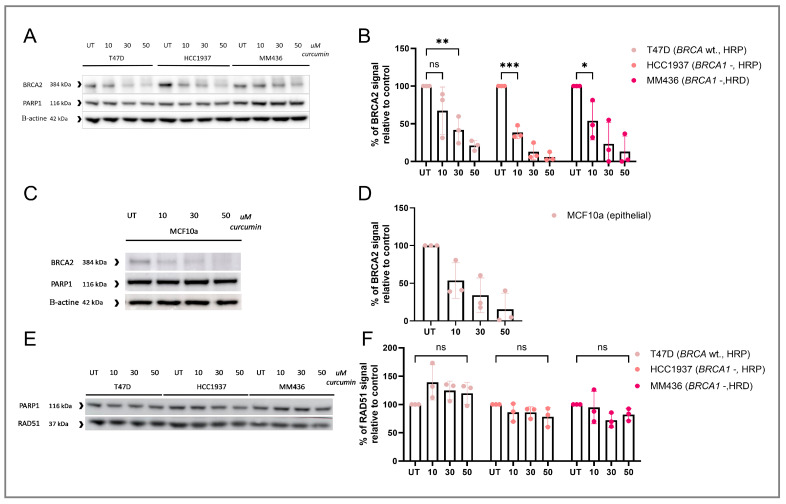
The effect of short curcumin treatment on BRCA2 and RAD51 levels. (**A**) BRCA2 Western blot after curcumin treatment of T47D (BRCA1 and 2 wt., HRP), HCC1937 (BRCA1-/-, HRP) and MM436 (BRCA1-/-, HRD). (**B**) Quantification of the BRCA2 Western Blot results from panel A, with correction to the loading control, with each data point representing the relative intensity of the BRCA2 band of each replicate and error bars representing the SEM (*n* = 3). (**C**) BRCA2 Western blot after curcumin treatment of MCF10a. (**D**) Quantification of the BRCA2 Western Blot results from panel C with correction to the loading control, with each data point representing the relative intensity of the BRCA2 band of each replicate and error bars representing the SEM (*n* = 3). (**E**) RAD51 Western blot after curcumin treatment of T47D (BRCA1 and 2 wt., HRP), HCC1937 (BRCA1-/-, HRP) and MM436 (BRCA1-/-, HRD). (**F**) Quantification of the RAD51 Western Blot results from panel E with correction to the loading control, with each data point representing the relative intensity of the BRCA2 band of each replicate and error bars representing the SEM (*n* = 3). Significant differences are indicated in graph B, D and F based on a standard one-way ANOVA test. Asterisks represent the significance based on the *p*-value, as follows: single (*) *p* ≤ 0.05, double (**) *p* ≤ 0.01, triple (***) *p* ≤ 0.001.

**Figure 4 cancers-17-02109-f004:**
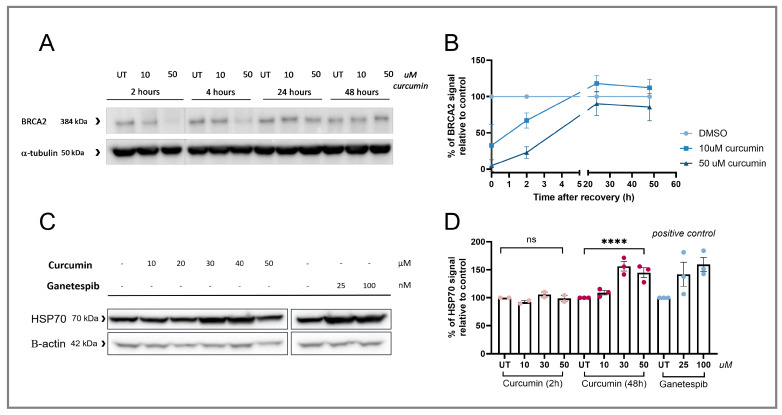
The effect of longer curcumin treatment on BRCA2 and HSP70 levels. (**A**) BRCA2 Western Blot results after curcumin treatment for 2 h and subsequent removal of the compound to allow recovery for 2, 24, and 48 h. (**B**) Quantification of the BRCA2 Western blot data from panel G with correction to the loading control, with each data point representing the relative intensity of the BRCA2 band of each replicate and error bars representing the SEM (*n* = 3). (**C**) HSP70 Western blot after curcumin and ganetespib treatment of T47D (BRCA1 and 2 wt., HRP). (**D**) Quantification of the HSP70 Western Blot results from panel I, with correction to the loading control, with each data point representing the relative intensity of the BRCA2 band of each replicate and error bars representing the SEM (*n* = 3). Significant differences are indicated in the graph based on a standard one-way ANOVA test. Asterisks represent the significance based on the *p*-value, as follows: quadruple (****) *p* ≤ 0.0001.

**Figure 5 cancers-17-02109-f005:**
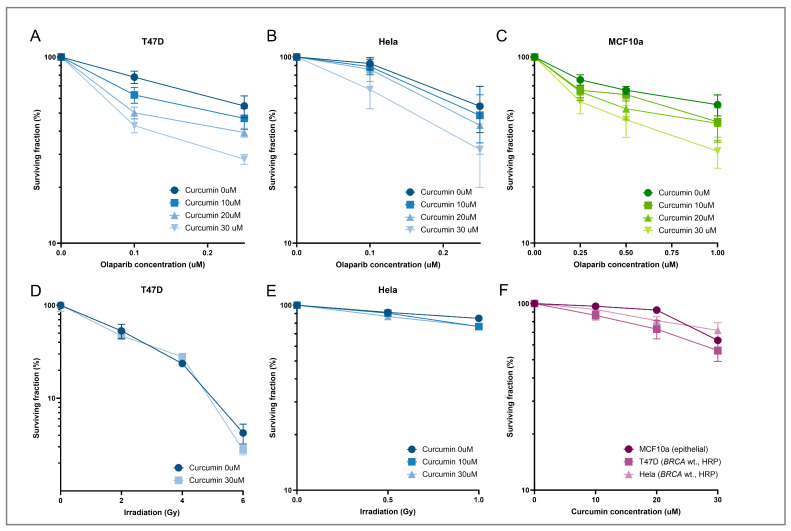
Clonogenic survival after combinatory treatment with curcumin and olaparib. (**A**–**C**) The results of the clonogenic survival assay of the T47D (**A**), HeLa (**B**), and MCF10a (**C**) cell lines after treatment with 10, 20, and 30 µM curcumin in combination with 0.1–1 µM olaparib. The surviving fraction is depicted in the graph with error bars indicating the SEM. (**D**,**E**) Clonogenic survival assay of T47D (**D**) and HeLa cells (**E**) after treatment with 10, 20, and 30 µM curcumin combined with 0–2 Gy X-irradiation. The surviving fraction is depicted in the graph with error bars indicating the SD. (**F**) The results of the clonogenic survival assay for the T47D, HeLa, and MCF10a cell lines after treatment with curcumin (10–30 µM) for 2.5 h.

## Data Availability

The authors declare that the data supporting the findings of this study are available within the paper, its Appendix A Files, and from the corresponding author upon reasonable request.

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
