# Peer review of "Curcumin Induces Homologous Recombination Deficiency by BRCA2 Degradation in Breast Cancer and Normal Cells"

_cancers, 2025, doi:10.3390/cancers17132109_

Round 1
Reviewer 1 Report
Comments and Suggestions for Authors
The authors caution that usage of curcumin during therapy for breast cancer could yield additional vulnerability by lowering the ability of non-cancerous cells to perform homologous recombination by degradation of BRAC2. More normal cells, might also be sensitive to trapping by PARP inhibitors playing into the HR deficiency. The transient nature of the effect does raise an issue over how significant this is. It would have been good to show whether there were long-term effects on the MCF10A cells after repeated curcumin exposure plus and minus PARP inhibitor. On lines 343-355, I wouldn't raise the issue of hemolysis specifically in the context of the paper, but as an aside-one more reason to be cautious. More relevant is the lack radiological response, although this needs to be expanded so more clearly relate to the context of the paper-was an effect expected?; was kind, etc.
Author Response
Dear Reviewer,
Thank you for taking the time to provide valuable feedback on our manuscript, titled “Curcumin induces homologous recombination deficiency by BRCA2 degradation in breast cancer and normal cells” by Zofia M. Komar et al., submitted to Cancers. We greatly appreciate your insightful comments and have made every effort to address them thoroughly, enhancing the manuscript to better align with the standards of Cancers.
To provide a clear explanation of the revisions made to the paper, we will address each of the comments individually:
“The transient nature of the effect does raise an issue over how significant this is.”
We appreciate this important observation. In a clinical context, curcumin supplementation would typically be administered daily, suggesting a repeated and potentially sustained biological effect. This implies that the transient reduction in BRCA2 and consequent homologous recombination (HR) deficiency might become significant with chronic exposure. Therefore, the potential sensitization of normal cells to PARP inhibitors (PARPi) could be more pronounced under conditions of continuous curcumin intake.
“It would have been good to show whether there were long-term effects on the MCF10A cells after repeated curcumin exposure plus and minus PARP inhibitor.”
We agree that this would indeed be a valuable extension of our work. A long-term exposure experiment in MCF10A cells, simulating chronic administration of both curcumin and PARPi, would provide insight into potential cumulative effects on normal epithelial cells. However, it is outside the scope of this paper.
“On lines 343-355, I wouldn't raise the issue of hemolysis specifically in the context of the paper, but as an aside-one more reason to be cautious.”
Thank you for this helpful suggestion. We agree that hemolysis, while not directly relevant to our experimental findings, supports the general need for caution when considering curcumin supplementation during therapy. We have adjusted the text accordingly and included a brief mention of this as an additional reason for careful consideration, rather than as a specific result of our study (see revised text on lines 357-358).
“More relevant is the lack radiological response, although this needs to be expanded so more clearly relate to the context of the paper-was an effect expected?; what kind, etc”
This is an excellent point. Without further explanation, the lack of response to ionizing radiation (IR) in our system may be unclear. Based on the observed decrease in RAD51 foci and associated BRCA2 degradation, we expected curcumin to inhibit homologous recombination. Ionizing radiation induces double-strand breaks, which are mostly repaired through the non-homologous end-joining (NHEJ) pathway. We included this experiment to check whether NHEJ still functions in the presence of curcumin. Since curcumin-treated cells did not show increased sensitivity to IR, this suggests that NHEJ remains active and curcumin mainly affects the HR pathway. This interpretation is supported by the observed sensitization to PARPi, where HR is more critical for repair. We have clarified this point in the revised Results section (lines 277-278).
Reviewer 2 Report
Comments and Suggestions for Authors
The manuscript addresses a timely and clinically relevant topic by investigating the effects of curcumin on homologous recombination (HR) and PARP inhibitor (PARPi) sensitivity in both cancerous and non-cancerous cells. The study is methodologically sound, well-structured, and provides valuable insights into curcumin’s potential as an HRD-inducing agent. However, several points require clarification and refinement:
-
Scientific Precision and Clarity
-
Please clarify the mechanistic link between BRCA2 degradation and HSP90 inhibition. While your hypothesis is reasonable, definitive causality remains unclear and should be presented as such.
-
The use of the term “PAIN compound” is appropriate, but its implications for curcumin’s therapeutic specificity and safety should be more explicitly discussed.
-
-
Impact on Normal Cells
-
The observation that BRCA2 suppression also occurs in normal cells (e.g., MCF10a) despite limited cytotoxicity is important. Please highlight this distinction more clearly to avoid misinterpretation.
-
Statistical replicates (e.g., n = 1 for HUVEC/C5RO) should be acknowledged as a limitation when interpreting normal cell data.
-
-
In Vitro vs. In Vivo Relevance
-
The conclusion that “relevant concentrations of curcumin are likely to induce toxicity” may be too strong based on currently available clinical data. Please rephrase to reflect that this is a potential risk, especially when using high-bioavailability formulations.
-
Consider summarizing the translational implications of your findings in a more structured and concise paragraph within the Discussion section.
-
-
Language and Style
-
Some sentences are overly long or repetitive (e.g., "should be approached with caution"). Consider editorial revision for conciseness and clarity.
-
Maintain consistency in terminology (e.g., “RAD51 foci” vs. “RAD51 IRIF”) and abbreviations throughout the manuscript.
-
Overall, your study is of high interest and could make a valuable contribution to the field after minor revisions. We encourage careful revision to improve clarity and strengthen the translational impact.
Comments on the Quality of English LanguageThe manuscript is generally written in clear and understandable English. However, several sentences would benefit from improved clarity and conciseness. In particular, there are some long or redundant phrases, inconsistent terminology (e.g., “RAD51 foci” vs. “RAD51 IRIF”), and occasional grammatical issues (e.g., unnecessary commas, article usage). Minor editorial revision by a native English speaker or professional editing service is recommended to enhance readability and ensure consistency throughout the text.
Author Response
Dear Reviewer,
Thank you for taking the time to provide valuable feedback on our manuscript, titled “Curcumin induces homologous recombination deficiency by BRCA2 degradation in breast cancer and normal cells” by Zofia M. Komar et al., submitted to Cancers. We greatly appreciate your insightful comments and have made every effort to address them thoroughly, enhancing the manuscript to better align with the standards of Cancers.
To provide a clear explanation of the revisions made to the paper, we will address each of the comments individually:
“Please clarify the mechanistic link between BRCA2 degradation and HSP90 inhibition. While your hypothesis is reasonable, definitive causality remains unclear and should be presented as such.”
Thank you for this important remark. In our original version, we had not explicitly linked HSP90 inhibition to BRCA2 degradation. Based on your feedback, we have now clearly stated that we hypothesize HSP90 inhibition to be the underlying cause of the observed BRCA2 degradation. While we acknowledge that definitive causality cannot be established within the scope of this study, this mechanistic connection is now explicitly described to clarify our interpretation (lines 308-309)
“The use of the term “PAIN compound” is appropriate, but its implications for curcumin’s therapeutic specificity and safety should be more explicitly discussed.”
Thank you for this valuable suggestion. Although the implications of curcumin being a PAIN compound were previously addressed across multiple parts of the discussion, we agree that the section where the term is introduced should more directly reflect its therapeutic relevance. We have now added a dedicated section discussing these implications in more detail (line 312-315).
“The observation that BRCA2 suppression also occurs in normal cells (e.g., MCF10a) despite limited cytotoxicity is important. Please highlight this distinction more clearly to avoid misinterpretation.”
We appreciate this insightful comment. Upon revising the relevant paragraph, we recognized the potential for misinterpretation. We have extended the conclusions to more clearly differentiate between the effects observed in cancer versus normal cells and to emphasize the combinatorial effects of curcumin and PARP inhibition. Additionally, we clarified that the limited toxicity was observed following curcumin monotherapy across all tested cell lines. These changes are reflected in lines 325-330.
“Statistical replicates (e.g., n = 1 for HUVEC/C5RO) should be acknowledged as a limitation when interpreting normal cell data.”
Thank you for pointing this out. We consider biological triplicates essential and have applied them consistently throughout the study, including for the MCF10A normal cell line. While the two additional normal lines were assessed once, they were used mostly to support the same trend, reinforcing the robustness of our findings.
“The conclusion that “relevant concentrations of curcumin are likely to induce toxicity” may be too strong based on currently available clinical data. Please rephrase to reflect that this is a potential risk, especially when using high-bioavailability formulations.”
A very important point—thank you. We have revised the concluding paragraph (lines 372-374) to explain this more cautiously, indicating that the findings suggest a potential risk, particularly with formulations that yield higher systemic concentrations, rather than a definitive toxicity.
“Consider summarizing the translational implications of your findings in a more structured and concise paragraph within the Discussion section. “
We appreciate this suggestion. Given that a substantial portion of the discussion is already dedicated to curcumin’s pharmacokinetics and clinical relevance (including patient plasma concentrations), we felt that adding a separate paragraph might lead to unnecessary repetition. Instead, we have restructured the concluding section to more clearly and concisely emphasize the translational implications of our findings.
Language revisions
We have conducted an additional language review to improve grammar and clarity throughout the manuscript. The use of abbreviations has also been standardized, with “RAD51 IRIF” now used consistently.